# Polymer-Nanodiscs as a Novel Alignment Medium for High-Resolution NMR-Based Structural Studies of Nucleic Acids

**DOI:** 10.3390/biom12111628

**Published:** 2022-11-03

**Authors:** Bankala Krishnarjuna, Thirupathi Ravula, Edgar M. Faison, Marco Tonelli, Qi Zhang, Ayyalusamy Ramamoorthy

**Affiliations:** 1Biophysics Program, Department of Chemistry, Biomedical Engineering, and Macromolecular Science and Engineering, Michigan Neuroscience Institute, University of Michigan, Ann Arbor, MI 48109, USA; 2National Magnetic Resonance Facility at Madison, University of Wisconsin-Madison, Madison, WI 53706, USA; 3Department of Biochemistry and Biophysics, University of North Carolina at Chapel Hill, Chapel Hill, NC 27599, USA

**Keywords:** nanodiscs, NMR, magnetic alignment, RNA, residual dipolar couplings

## Abstract

Residual dipolar couplings (RDCs) are increasingly used for high-throughput NMR-based structural studies and to provide long-range angular constraints to validate and refine structures of various molecules determined by X-ray crystallography and NMR spectroscopy. RDCs of a given molecule can be measured in an anisotropic environment that aligns in an external magnetic field. Here, we demonstrate the first application of polymer-based nanodiscs for the measurement of RDCs from nucleic acids. Polymer-based nanodiscs prepared using negatively charged SMA-EA polymer and zwitterionic DMPC lipids were characterized by size-exclusion chromatography, ^1^H NMR, dynamic light-scattering, and ^2^H NMR. The magnetically aligned polymer-nanodiscs were used as an alignment medium to measure RDCs from a ^13^C/^15^N-labeled fluoride riboswitch aptamer using 2D ARTSY-HSQC NMR experiments. The results showed that the alignment of nanodiscs is stable for nucleic acids and nanodisc-induced RDCs fit well with the previously determined solution structure of the riboswitch. These results demonstrate that SMA-EA-based lipid-nanodiscs can be used as a stable alignment medium for high-resolution structural and dynamical studies of nucleic acids, and they can also be applicable to study various other biomolecules and small molecules in general.

## 1. Introduction

Residual dipolar couplings (RDCs), providing long-range angular information within a molecule, are increasingly used to measure atomic-resolution structural and dynamic features for a variety of molecules [1,2,3,4]. They are also useful in validating and refining biomolecular structures determined by X-ray crystallography and nuclear magnetic resonance (NMR) spectroscopy [5]. RDCs are through-space interactions; however, unlike NOEs (1–5 Å) and PREs (10–30 Å), they report on structural information irrespective of distance since RDCs arise from orientations between two magnetically active nuclei with respect to a strong, uniform magnetic field present throughout the sample [6]. However, when molecules tumble isotropically in solution conditions, dipolar couplings are averaged to zero and render such structural information inaccessible. Therefore, a special medium that can partially restrict a molecule’s isotropic tumbling and induce weak alignment in an external magnetic field is needed to capture residuals of dipolar couplings [7]. Various alignment media, including bicelles [8,9,10,11], Pf1 phage [6,12,13,14,15,16], flagella [17], constrained gels (polyacrylonitrile) [18], graphene oxide grafted with polymer brushes [19], and most recently, polymer-based lipid-nanodiscs (polymer-nanodiscs) [20,21], have been used to measure RDCs for various biomolecules [1,2,22,23]. Although bicelles are commonly used as an alignment medium in NMR studies [24,25,26,27,28,29], they can be unstable due to detergents [22]. In contrast, polymer-nanodiscs, similar to protein/peptide-based nanodiscs [30,31,32,33,34,35,36,37,38,39], comprise a planar lipid bilayer surrounded by a polymer belt and can be stable due to a detergent-free environment [40,41]. The ability to use aligned and 90°-flipped polymer-based nanodiscs is another unique advantage [21]. In addition, polymer-nanodiscs are stable against a broad range of pH conditions and resistant to aggregation by divalent metal ions [42,43,44].

Previous studies have demonstrated that a transition from the gel (isotropic) to liquid–crystalline (anisotropic) phase, when the temperature is raised above the phase transition temperature (T_m_) of lipids, renders magnetic alignment of nanodiscs such that the bilayer normal is oriented perpendicular to the external magnetic field [38,42,45,46,47,48,49,50]; in addition, a recent solid-state NMR study demonstrated the magnetic alignment of peptoid-based nanodiscs [51]. When water-soluble molecules are placed in liquid-crystalline nanodiscs, their motion is partially restricted due to the anisotropic environment induced by the aligned nanodiscs [21]. Hence, the polymer-nanodiscs can be used to measure RDCs for various molecules, including small molecules, proteins, and nucleic acids. We have previously reported the use of polymer-nanodiscs for RDCs measurement from water-soluble proteins [20,21]. Here, we report the first demonstration of the application of styrene-maleic acid-ethylamine (SMA-EA)-based 1,2-dimyristoyl-sn-glycero-3-phosphocholine (DMPC)-nanodiscs for RDC measurement from the *Bacillus cereus* fluoride riboswitch aptamer, a non-coding RNA that detects cellular fluoride and activates gene expression involved in fluoride toxicity response [52,53]. The measured RDCs are in excellent agreement with the previously determined solution structure of the riboswitch [54], indicating the suitability of polymer-nanodiscs for high-resolution structural studies of nucleic acids.

## 2. Materials and Methods

### 2.1. Preparation of DMPC Liposomes

A total of 120 mg of DMPC (Avanti Polar Lipids; AL, USA) was taken in a 15 mL centrifuge tube and dissolved in a solvent mixture containing 1:1 *v*/*v* CH_3_OH:CHCl_3_ (600 µL each) (Sigma-Aldrich; St. Louis, MO, USA). The organic solvents were evaporated by applying a low-pressure N_2_ gas (20 min) onto the lipid–solvents mixture. (Caution: N_2_ gas at high pressure can spill the sample out of the centrifuge tube). The lipid mixture was then subjected to a vacuum for 1 h to remove all the residual solvents. Finally, the liposome solution was prepared by resuspending the solvent-free lipid mixture in a 10 mM potassium phosphate buffer (pH 7.4) containing 50 mM NaCl, and by subjecting it to the freeze–thaw cycle three times (using liquid N_2_ and hot water (~70 °C)) and vortexing for 5 s.

### 2.2. SMA-EA Polymer

The SMA-EA polymer was synthesized, purified, and characterized using the published protocol [42].

### 2.3. Preparation of Polymer-Nanodiscs

The SMA-EA stock solution was prepared by dissolving the lyophilized polymer in a 10 mM potassium phosphate buffer (pH 7.4) containing 50 mM NaCl at 100 mg/mL concentration. The pH of the solution was checked and adjusted to ~7.4 by adding HCl before using it in nanodisc preparation.

The liposome solution was mixed with SMA-EA solution at a 1:1 (*w*/*w*) lipid: polymer ratio and incubated at 30 °C overnight. Then, the nanodiscs solution was purified by size-exclusion chromatography (SEC) using fast protein liquid chromatography (FPLC; GE Healthcare, Chicago, IL, USA) and 10 × 300 Superdex 200 column (GE Healthcare). The fractions (detected at 254 nm) containing polymer-nanodiscs were combined and concentrated to 100 mg DMPC/mL using a 100 kDa cut-off Amicon Centricon membrane filter (Burlington, MA, USA). The polymer-nanodiscs were stored under −20 °C. When needed, the nanodiscs sample was warmed up to room temperature before the addition of RNA and used for NMR measurements.

### 2.4. Characterization of SMA-EA Nanodiscs

#### 2.4.1. ^1^H NMR

The 1D ^1^H and 2D ^1^H/^1^H NOESY NMR spectra were recorded on a Bruker 500 MHz NMR spectrometer at 298 K. NMR spectra of SEC-purified polymer-nanodiscs (before concentrating the sample for RDC measurements) were recorded in a 10 mM potassium phosphate buffer (pH 7.4) containing 50 mM NaCl, and processed using Topspin (version 3.6.2, Bruker, Billerica, MA, USA) and calibrated to water proton peak.

#### 2.4.2. Dynamic Light Scattering (DLS)

DLS experiments were performed at 25 °C using Wyatt Technology^®^, DynaPro^®^, NanoStar^®^, and a 1 µL quartz MicroCuvette. The DLS data were collected on the SEC purified polymer-nanodiscs.

### 2.5. NMR Sample Preparation

Uniformly ^13^C- and ^15^N-labeled fluoride riboswitch aptamer samples were prepared as described previously [54]. Briefly, samples were transcribed in vitro using T7 polymerase based on DNA purchased from Integrated DNA Technologies (IDT), Inc., (Coralville, IA, USA). The transcribed samples were ethanol precipitated and gel-purified under denaturing conditions, eluted using the Elutrap system (Whatman), and subsequently purified using a Hi-Trap Q column (GE Healthcare). The purified samples were exchanged to 10 mM sodium phosphate (pH 6.4) (containing 50 mM KCl, 2 mM MgCl_2_, 10 mM NaF, and 50 μM EDTA) and concentrated to ~0.5–1.0 mM using Amicon filtration systems (Millipore). For preparing samples for residual dipolar coupling (RDC) measurements, the lyophilized powder of ^13^C- and ^15^N-isotope-labeled RNA was first dissolved in 100 µL of a 10 mM potassium phosphate buffer (pH 7.4, containing 50 mM NaCl) and followed by the addition of 180 µL of polymer-nanodiscs solution (100 mg/mL of lipids). A total of 10% of ^2^H_2_O was included in all NMR samples.

### 2.6. RDC NMR Measurements

The NMR experiments for RDC measurements were carried out on a Varian VNMRS 600 MHz (^1^H) spectrometer equipped with a H/C/N cryogenically cooled probe. The NMR experiments were recorded in isotropic (288 K) and anisotropic (308 K) conditions. RDCs were extracted using 2D ARTSY-HSQC experiments with the reference and attenuated spectra were recorded in an interleaved fashion [55]. The amplitude-encoding ARTSY delay was optimized empirically for each experiment and is reported in Appendix A, along with acquisition parameters. After recording the spectra in the isotropic condition with the sample at 288 K, the probe temperature was increased to 308 K to allow the alignment of nanodiscs in the magnetic field. In addition, 1D deuterium NMR spectra were recorded using the same sample before and after acquiring the 2D ARTSY-HSQC data.

RDCs for the H1-N1 and H3-N3 imino groups were recorded using an ARTSY BEST-HSQC pulse program that uses band-selective ^1^H pulses and leaves the water magnetization unperturbed [55]. RDCs for H6-C6, H8-C8, H2-C2 aromatic groups, and H5-C5 and H1′-C1′ groups were recorded using two separate ARTSY-HSQC experiments that employ band-selective ^13^C pulses to allow for the removal of homonuclear ^13^C-^13^C couplings in the ^13^C-dimension. RDCs for the remaining ribose groups were obtained using an ARTSY-HSQC experiment with constant-time ^13^C-evolution to remove homonuclear ^13^C-^13^C couplings, with the constant-time delay set to 28.6 ms. NMR spectra were processed using NMRPipe [56].

### 2.7. Measuring Nanodisc-Induced RDCs in Holo Fluoride Riboswitch

Two-dimensional NMR spectra were viewed and analyzed using NMRFAM-Sparky v3.190 [57]. Assigned peak intensities for isotropic reference (*I_IR_*), isotropic attenuated (*I_IA_*), anisotropic reference (*I_AR_*), and anisotropic attenuated (*I_AA_*) spectra per chemical group set (C2H2/C6H6/C8H8, C1′C1′/C5H5, N1H1/N3H3) were extracted. Intensity values were used to calculate RDC values (*D*) according to the following Equation (1):(1)D=±2πT(cos−1IAAIAR2−cos−1IIAIIR2)
where the mixing time, *T*, was 5.0 ms for C2H2/C6H6/C8H8, 6.0 ms for C1′H1′/C5H5, and 11.5 ms for N1H1/N3H3. Values from severely overlapped peaks were omitted from further analysis, resulting in a total of 84 nanodisc-induced measured RDCs. A subset of total RDC values (*N* = 73), omitting the values from the flexible region J1/2 (residues A17, U18, A19, A20, A21, C22), was fit to the lowest energy structure (PDB structure 5KH8) using the program RAMAH [58] to generate the alignment tensor, back-calculate RDCs, and quality (*Q*) factors [59]. Back-calculated RDCs per alignment medium set were plotted against measured RDCs for both nanodisc- and phage-induced sets. Errors for nanodisc and phage RDCs were estimated from RAMAH, scaling approximately to the range of each RDC set (4 Hz for nanodisc, 2 Hz for phage).

## 3. Results and Discussion

Since RNA is a polyanion, negatively charged SMA-EA-based DMPC-nanodiscs were used to avoid non-specific coulombic interactions between RNA and the polymer belt of the nanodisc [60,61]. The polymer-nanodiscs were prepared using SMA-EA polymer and DMPC at a 1:1 (*w*/*w*) ratio. The turbid DMPC liposome solution turned transparent after mixing and incubating with SMA-EA overnight at ~30 °C, suggesting the formation of aqueous soluble polymer-nanodiscs (Figure 1A). The polymer-nanodiscs sample was then purified by SEC. The SEC chromatogram showed two major peaks: the peak between 9–13.5 mL corresponds to polymer-nanodiscs, and the peak between 14–20.5 mL corresponds to excess-free SMA-EA polymer (Figure 1B). The single symmetric elution peak observed in SEC is an indication of size homogeneity for the nanodiscs. Nanodiscs with such size homogeneity are suitable for high-resolution structural studies of membrane proteins by cryo-EM and NMR spectroscopy. Polymer-nanodiscs were then characterized by ^1^H NMR under isotropic conditions (~15 mg/mL). The ^1^H NMR spectrum showed peaks from acyl-chain protons (0.3–0.7 ppm), characteristic quaternary ammonium protons (3.22 ppm) from DMPC lipids, and a broad peak from the aromatic styrene-ring of SMA-EA (6.4–7.6 ppm), indicating the presence of SMA-EA polymer and the DMPC lipids in the SEC-purified sample (Figure 1C). A 2D NOESY spectrum showed internuclear NOEs between the SMA-EA aromatic group and DMPC acyl-chains (Appendix A), confirming the formation of SMA-EA-DMPC assemblies. The size of polymer-nanodiscs was estimated by dynamic light scattering (DLS) at a DMPC concentration of 3 mg/mL, reporting a hydrodynamic radius of ~10.5 ± 1 nm (Figure 1C and Appendix A). The excess-free polymer in the sample (14–20.5 mL of SEC chromatogram (Figure 1B)) was confirmed by ^1^H NMR (Appendix A).

The lyophilized powder of fluoride-bound ^13^C/^15^N-labeled fluoride riboswitch aptamer was dissolved in a 10 mM potassium phosphate buffer (pH 7.4), 50 mM NaCl, and added to the SEC-purified SMA-EA nanodiscs sample. The magnetic alignment of nanodiscs was confirmed by deuterium NMR (Figure 2A). At 288 K (below the T_m_ of DMPC (~24 °C)), the deuterium NMR spectrum showed a single peak, indicating the isotropic nature of nanodiscs (Figure 2A (bottom) and Figure 2B). In contrast, at 308 K (above T_m_), a doublet with a residual quadrupolar coupling (RQC) value of ~8 Hz was observed, indicating the magnetic alignment of polymer-nanodiscs in a strong external magnetic field (Figure 2A (top) and Figure 2C).

To determine one-bond RDCs for different ^13^C-^1^H and ^15^N-^1^H pairs of nucleotides in fluoride riboswitch aptamer (Figure 3A,B), a series of 2D ARTSY-HSQC NMR experiments [55] (listed in Appendix A) were recorded at 288 K (isotropic) and 308 K (anisotropic). Each ARTSY spectrum was acquired twice in an interleaved manner, yielding two separate spectra after processing, where the first one corresponds to the reference spectrum and the second corresponds to the attenuated spectrum. In the attenuated spectrum, the intensity of peaks is modulated by the strength of the coupling (scalar, J, and dipolar, D). After data processing, J or J + D values were obtained by taking the ratio of reference spectra intensities over the attenuated spectra. Finally, the RDCs were obtained by subtracting the couplings measured under isotropic conditions from the anisotropic conditions. Previously reported NMR assignments for the fluoride-bound state [54] were used to assign the cross-peaks in ARTSY spectra (Figure 3C–E, Appendix A and Appendix A). The RDC values for adenine (C2-H2, C8-H8), guanine (N1-H1, C8-H8), uridine (N3-H3, C5-H5, C6-H6), cytosine (C5-H5, C6-H6), and ribose (C1′-H1′) were extracted using an RD tool in NMRFAM-SPARKY software (Figure 3C–E). In addition, the alignment of polymer-nanodiscs was found to be stable as the ^2^H NMR spectrum recorded 5 days after the sample preparation showed no change in the RQC value (~8.0 Hz) (Figure 2D). Therefore, the polymer-nanodiscs were stable during the acquisition of all 2D ARTSY-HSQC NMR experiments for the RDC measurements.

A total of 84 RDCs were measured for the fluoride-bound fluoride riboswitch aptamer in polymer-nanodiscs, where severely overlapped peaks were omitted (Figure 4A,B). It was previously shown that the fluoride riboswitch aptamer adopts identical solution structures in the presence and absence of fluoride, where an excellent correlation was observed between RDCs measured in apo and holo riboswitches with Pf1 phage as the alignment medium [54]. To evaluate the quality of the nanodisc-induced RDCs, we employed the program RAMAH [58] and carried out order tensor analysis using the apo- fluoride riboswitch aptamer structure (PDB: 5KH8) [54]. Here, a subtotal of 73 nanodisc-induced RDCs was used, where RDCs from the flexible region J12 (Figure 4A; residues A17, U18, A19, A20, A21, C22) were excluded. An excellent agreement between the measured and back-calculated RDCs was observed with a quality (*Q*) [59] factor = 14.98% (Figure 4C), which is consistent with a similar RAMAH analysis of previously measured Pf1-phage-induced RDCs of the holo state [54] with *Q* factor = 14.79% (Figure 4D). A direct comparison between shared RDCs measured from polymer-nanodiscs and those from Pf1 phage (*N* = 73; including J12) revealed an anti-correlation with a correlation coefficient *R*^2^ of 0.97 (Figure 4E). Indeed, order tensor analyses of these two sets of RDCs showed that the nanodiscs-induced alignment is parallel to the magnetic field (Szz = +1.25 ± 0.01 × 10^−3^), whereas the Pf1-phage-induced alignment is perpendicular to the magnetic field (Szz = −0.77 ± 0.01 × 10^−3^). Overall, the measured RDCs from polymer-nanodiscs are in excellent agreement with the RDCs back-calculated from the structure and with similar accuracy to those measured from the Pf1 phage medium. This finding successfully demonstrates the suitability of polymer-nanodiscs as an alignment medium for structural studies on RNA and the reproducibility of these measurements across different media [54]. The results also suggest that it is very unlikely that the negatively charged RNA interacts with either negatively charged SMA-EA or zwitterionic DMPC lipids.

It is important to note that the charge of the polymer belt should be the same as the net charge of the molecule to be studied at a given pH to avoid any non-specific electrostatic interactions [20,21,60]. Therefore, nanodiscs prepared using a charged polymer cannot be used to study protein–protein and protein–nucleic acid complexes that are stabilized by electrostatic interactions [62]. On the other hand, non-ionic polymer-nanodiscs can be suitable for studying such complexes constituting differently charged molecules, as demonstrated for the membrane-bound redox complex composed of cationic cytochrome P450 (CYP450) and anionic CYP450 reductase [63]. Another advantage is that the non-ionic polymers do not absorb light in the UV region; therefore, they do not interfere with the UV-based characterization of biomolecules unlike the SMA-based polymers [64]. We also note that zwitterionic polymers may be another alternative to characterize biomolecules with different net charges [65]. Previous studies have shown that magnetically aligned polymer-nanodiscs are excellent tools for high-resolution NMR-based structural studies of membrane proteins and cytosolic proteins [20,21,66]. Together, the results indicate the power of polymer-nanodiscs to extract RDCs for high-resolution structure determination of a wide range of biomolecules and to validate the reported biomolecular structures determined by X-ray crystallography and NOE-based NMR studies.

## 4. Conclusions

In conclusion, this study demonstrated the application of polymer-nanodiscs as a new weak-alignment medium to measure RDCs for nucleic acids using NMR experiments. It has been shown recently that RNA can adopt alternative conformations upon protonation to direct biological outcomes [67]. The stability of polymer-nanodiscs at low pHs [42,43,44] would enable measuring RDCs as high-resolution structural constraints for pronated RNAs that would be difficult to obtain using other widely used alignment media for nucleic acids, such as Pf1 phage [68]. With their long-term stability, as indicated by deuterium NMR, the polymer-nanodiscs can be used to record multi-dimensional biomolecular NMR experiments that require longer-acquisition times. Furthermore, due to lipid-solubilizing properties, SMA-EA can be used to directly isolate membrane proteins along with native lipids [61,69], and its compatibility with divalent metal ions makes it suitable for studying metalloproteins and metal-bound nucleic acids. Thus, the polymer-nanodiscs are excellent tools for studying the conformation of membrane proteins, soluble proteins, and nucleic acids (current study) and can even extend to characterize natural products and small-molecule drug candidates in drug-development studies [70,71,72,73].

## 5. Patents

The authors declare that the SMA-EA polymer is US patented.

## Figures and Tables

**Figure 1 biomolecules-12-01628-f001:**
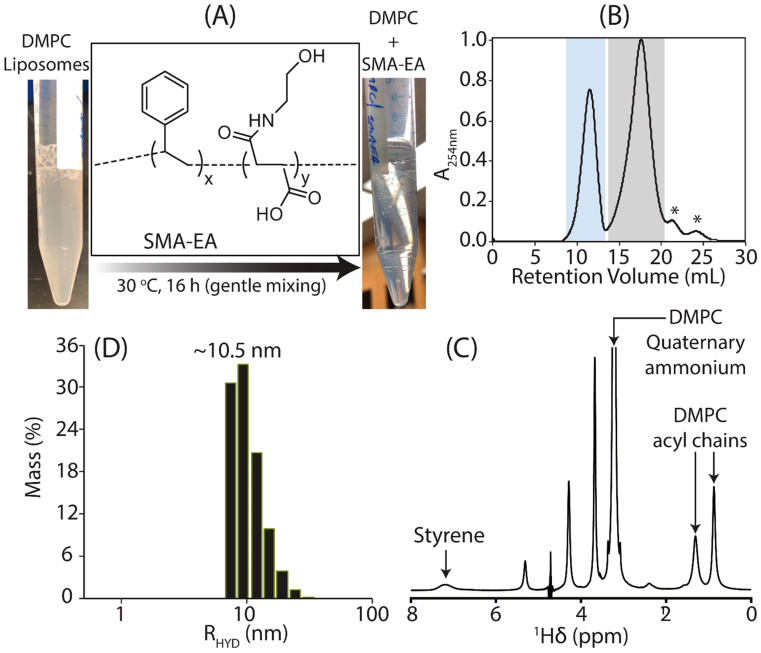
(**A**) An image of DMPC liposome solution without (left-side; turbid solution) and with (right-side; transparent solution) SMA-EA polymer; SMA-EA chemical structure is shown. (**B**) SEC chromatogram of the purification of polymer-nanodiscs prepared using a 1:1 (*w*/*w*) polymer: lipid ratio. The elution peaks highlighted in blue (9–13.5 mL) and gray boxes (14–20.5 mL) are from polymer-nanodiscs and free polymer, respectively. The fractions covered within each box were combined before characterization. ‘*’ indicates the uncharacterized peaks. (**C**) ^1^H NMR spectrum showing the characteristic peaks from DMPC and polymer labeled with assignments. The spectrum was recorded at 298 K with a DMPC concentration of ~15 mg/mL. (**D**) DLS profile of polymer-nanodiscs showing a hydrodynamic radius of ~10.5 ± 1 nm recorded at 32 °C using a ~3 mg/mL concentration of polymer-nanodiscs.

**Figure 2 biomolecules-12-01628-f002:**
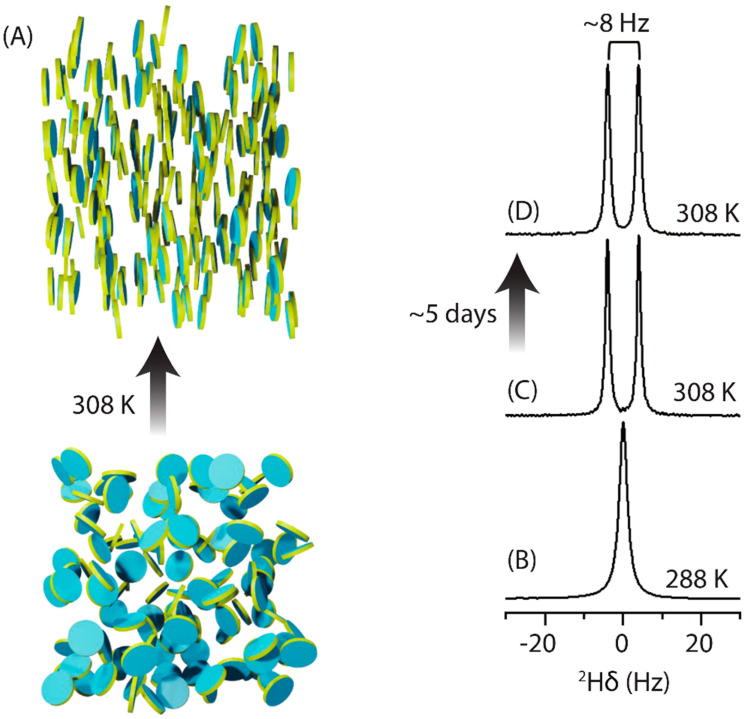
Magnetic alignment of polymer-nanodiscs in an external magnetic field. (**A**) Schematic of the temperature-dependent isotropic (bottom) and magnetic alignment (top) of polymer-nanodiscs. (**B**–**D**) Deuterium NMR spectra of the polymer-nanodiscs at 288 K under isotropic condition (**B**), and at 308 K under aligned (day 1 (**C**) and day 5 (**D**)) conditions. The polymer-nanodisc sample was prepared using ~64 mg/mL DMPC concentration in 10 mM potassium phosphate buffer (pH 7.4) containing 50 mM NaCl, and the spectra were recorded using a Varian VNMRS 600 MHz NMR spectrometer.

**Figure 3 biomolecules-12-01628-f003:**
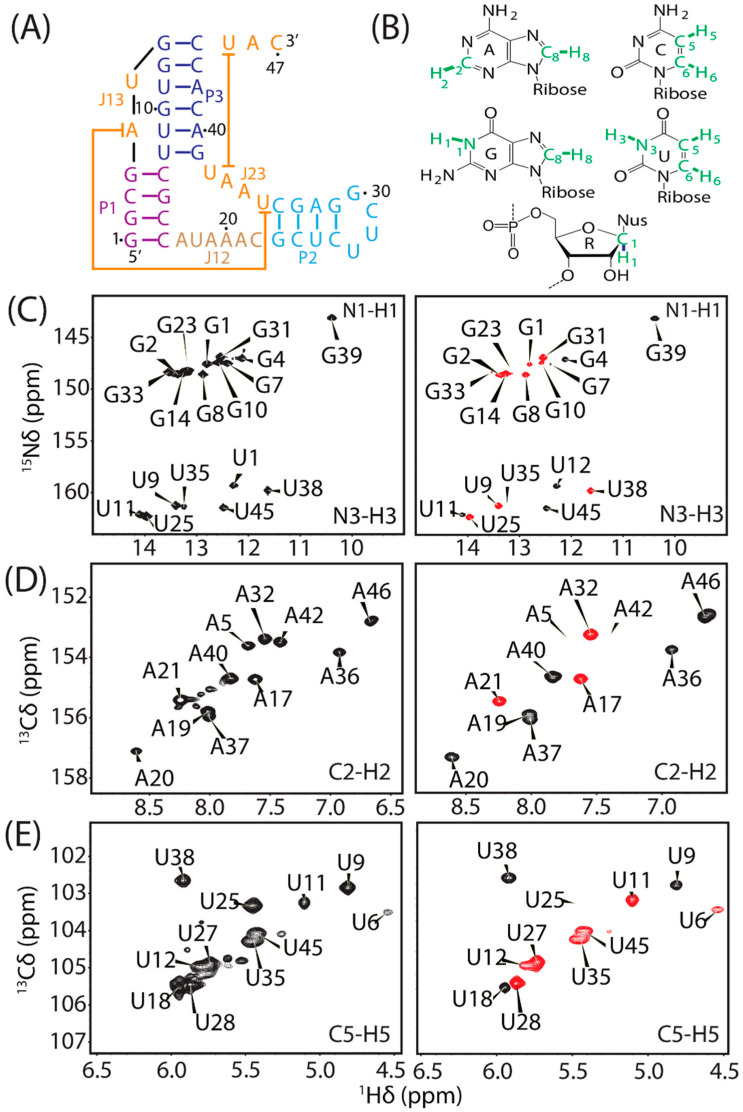
(**A**) Secondary structure of the fluoride riboswitch aptamer from *B. cereus,* color-coded by region. (**B**) Chemical structures of RNA nucleotide moieties. RDCs were measured for the highlighted (green) atom pairs. (**C**–**E**): Reference (left-side column) and attenuated (right column) 2D ARTSY-HSQC spectral regions displaying cross-peaks from (**C**) guanine N1-H1, uridine N3-H3; (**D**) adenine C2-H2; and (**E**) uridine C5-H5. Positive and negative contours are colored black and red, respectively. All NMR spectra were recorded in the presence of the magnetically aligned polymer-nanodiscs medium. The sample was made up of ~64 mg/mL DMPC in 10 mM potassium phosphate buffer (pH 7.4) and 50 mM NaCl. The NMR data were collected on a Varian VNMRS 600 MHz NMR spectrometer operated with the probe temperature of 308 K.

**Figure 4 biomolecules-12-01628-f004:**
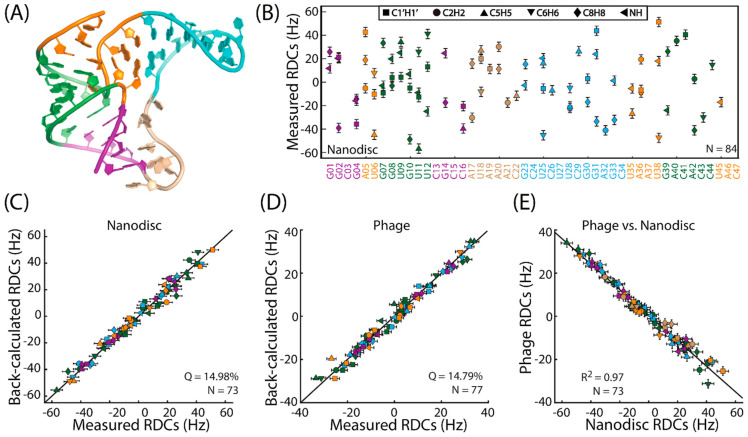
(**A**) Solution structure of the apo *B. cereus* fluoride riboswitch aptamer used in the NMR study (PDB: 5KH8) [54]. (**B**) Experimentally measured RDCs from polymer-nanodiscs alignment medium. (**C**,**D**) Correlation of experimentally measured and calculated RDCs for polymer-nanodiscs (**C**) and Pf1 phage (**D**) media. (**E**) Correlation of RDCs measured using polymer-nanodiscs and Pf1 phage alignment media.

## Data Availability

Not applicable.

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
