# Peer review of "Polymer-Nanodiscs as a Novel Alignment Medium for High-Resolution NMR-Based Structural Studies of Nucleic Acids"

_biomolecules, 2022, doi:10.3390/biom12111628_

Round 1

Reviewer 1 Report

The authors of the manuscript  untitled have  demonstrated for the first time the application of polymer-nanodiscs for the measurement of RDCs from nucleic acids. These nanodiscs can be a stable alignment medium for high-resolution structural and dynamical studies of nucleic acids, and they can also be applicable to studying various other biomolecules and small molecules in general. The nanodiscs have acquired a lot of the attention of the life-science community. The originality of results makes this manuscript appropriate for publication in MDPI Biomolecules.  The paper is of high scientific interest and could be published in the present form.

Author Response

We thank the reviewer for recommending our study for publication. 

Reviewer 2 Report

The manuscript “Polymer-nanodiscs as a novel alignment medium for high-resolution NMR based structural studies of nucleic acids” presents the finding that polymer-based nanodiscs can be used to collect residual dipolar couplings that are useful for calculation of accurate macromolecular structures including of RNA.  This is an important finding, as current alignment media are often unsuitable due to inherent limitations such as instability, while the polymer nanodiscs are thought to be more stable. The results will be of interest to structural biologists, membrane biophysicists and polymer chemists.  The text is well written and the figures are accessible, and only minor issues are noted below that could further improve the manuscript.

1.     The authors could show or discuss the long-term stability of the NMR samples containing the polymer-nanodiscs, as this is proposed to be a key advantage. We note that they do state that “In addition, polymer-based nanodiscs are stable against a broad range of pH conditions and resistant to aggregation by divalent metal ions. With their long-term stability, as indicated by deuterium NMR, the polymer-nanodiscs can be used to record multi-dimensional biomolecular NMR experiments that require longer-acquisition times”.  However this deuterium NMR data is not shown and the timeframes and storage conditions of the sample preparation and data collection is not presented.

2.     The specific disadvantages of the polymer-nanodiscs including heterogeneity in size should be discussed as well as the advantages to provide a balance perspective.

3.     Although the authors focus on the use of SMA-EA, other similar polymers have also been shown to be useful to align macromolecules such as proteins, including by Sang Ho Park et al, Membrane proteins in magnetically aligned phospholipid polymer discs for solid-state NMR spectroscopy. Such studies should also be referenced and discussed.

4.     The use of cationic and zwitterionic polymers to produce discs for alignment of macromolecules with various properties including charge should be discussed, as this could broaden the impact of the study.

Author Response

Reviewer 2

The manuscript “Polymer-nanodiscs as a novel alignment medium for high-resolution NMR based structural studies of nucleic acids” presents the finding that polymer-based nanodiscs can be used to collect residual dipolar couplings that are useful for calculation of accurate macromolecular structures including of RNA.  This is an important finding, as current alignment media are often unsuitable due to inherent limitations such as instability, while the polymer nanodiscs are thought to be more stable. The results will be of interest to structural biologists, membrane biophysicists and polymer chemists.  The text is well written and the figures are accessible, and only minor issues are noted below that could further improve the manuscript.

  1. The authors could show or discuss the long-term stability of the NMR samples containing the polymer-nanodiscs, as this is proposed to be a key advantage. We note that they do state that “In addition, polymer-based nanodiscs are stable against a broad range of pH conditions and resistant to aggregation by divalent metal ions. With their long-term stability, as indicated by deuterium NMR, the polymer-nanodiscs can be used to record multi-dimensional biomolecular NMR experiments that require longer-acquisition times”.  However this deuterium NMR data is not shown and the timeframes and storage conditions of the sample preparation and data collection is not presented.

Our Response: The 1D deuterium NMR spectra recorded at different time points are shown in Figure 2 (C, D).

Additionally, the following text is included in the manuscript.

“…the alignment of polymer-nanodiscs was found to be stable as the 2H NMR spectrum recorded 5-days after the sample preparation showed no change in the RQC value (~8.0 Hz) (Figure 2D). Therefore, the polymer-nanodiscs were stable during the acquisition of all 2D ARTSY-HSQC NMR experiments for the RDC measurements.”

The Material and Methods section is updated with the sample storage conditions as follows.

“In addition, 1D deuterium NMR spectra were recorded using the same sample before and after acquiring the 2D ARTSY-HSQC data.”

“The polymer-nanodiscs were stored under -20 °C. When needed, the nanodiscs sample was warmed up to room temperature before the addition of RNA and used for NMR measurements.”

We then recorded the spectra continuously without having to store the sample outside of the magnet.

  1. The specific disadvantages of the polymer-nanodiscs including heterogeneity in size should be discussed as well as the advantages to provide a balance perspective.

Our Response:

We agree with the reviewer.

Polymer nanodiscs are suitable for the functional and structural characterization of both membrane and soluble proteins. The SMA-EA nanodiscs used in this study are homogenous in size as indicated by the SEC and DLS profiles (Figure 1B, D). However, the main disadvantage of SMA-polymer nanodiscs is the charge on the polymer belt that interferes with the characterization of oppositely charged molecules. The other disadvantage is that the SMA polymers absorb light in the UV region, and can interfere with the UV-based protein measurements. This is discussed in the revised manuscript by including the following text.

“It is important to note that the charge of the polymer belt should be the same as the net charge of the molecule to be studied at a given pH to avoid any non-specific electrostatic interactions (ChemComm 2018, 54, 9615-9618, Langmuir 2022, 38, 244-252, Angew. Chem. Int. Ed. 2019, 58, 14925-14928). Therefore, nanodiscs prepared using a charged polymer cannot be used to study protein-protein and protein-nucleic acid complexes that are stabilized by electrostatic interactions (J. R. Soc. Interface. 2011, 8, 1065-1078). On the other hand, non-ionic polymer-nanodiscs can be suitable for studying such complexes constituting differently charged molecules, as demonstrated for the membrane bound redox-complex composed of cationic cytochrome P450 (CYP450) and anionic CYP450 reductase (Analytical Chemistry 2022, 94, 11908-11915). Another advantage is that the non-ionic polymers do not absorb light in the UV region; and, therefore, they do not interfere with UV-based characterization of biomolecules unlike the SMA based polymers (ChemComm 2022, 58, 4913-4916). We also note that zwitterionic polymers may be another alternative to characterize biomolecules with different net charges (Sci. Rep. 2017, 7, 7432).”

  1. Although the authors focus on the use of SMA-EA, other similar polymers have also been shown to be useful to align macromolecules such as proteins, including by Sang Ho Park et al, Membrane proteins in magnetically aligned phospholipid polymer discs for solid-state NMR spectroscopy. Such studies should also be referenced and discussed.

Our Response: The suggested references by Sang Ho Park et al are cited.

  1. The use of cationic and zwitterionic polymers to produce discs for alignment of macromolecules with various properties including charge should be discussed, as this could broaden the impact of the study.

Our Response: Agree.

To address this comment, the following text is added in the revised manuscript.

“It is important to note that the charge of the polymer belt should be the same as the net charge of the molecule to be studied at a given pH to avoid any non-specific electrostatic interactions (ChemComm 2018, 54, 9615-9618, Langmuir 2022, 38, 244-252, Angew. Chem. Int. Ed. 2019, 58, 14925-14928). Therefore, nanodiscs prepared using a charged polymer cannot be used to study protein-protein and protein-nucleic acid complexes that are stabilized by electrostatic interactions (J. R. Soc. Interface. 2011, 8, 1065-1078). On the other hand, non-ionic polymer-nanodiscs can be suitable for studying such complexes constituting differently charged molecules, as demonstrated for the membrane bound redox-complex composed of cationic cytochrome P450 (CYP450) and anionic CYP450 reductase (Analytical Chemistry 2022, 94, 11908-11915). Another advantage is that the non-ionic polymers do not absorb light in the UV region; and, therefore, they do not interfere with UV-based characterization of biomolecules unlike the SMA based polymers (ChemComm 2022, 58, 4913-4916). We also note that zwitterionic polymers may be another alternative to characterize biomolecules with different net charges (Sci. Rep. 2017, 7, 7432).”

Reviewer 3 Report

Prof. Ayyalusamy Ramamoorthy and co-workers previously studied the advanced solid-state NMR techniques of membrane proteins and peptides using various unique magnetic alignment systems such as nanodisc, polymer-nanodisc, bicelles, mechanically oriented media and so on (Biophys. J. 2002, JACS 2004, Anal. Chem. 2012, Sci. Rep. 2013, PNAS 2019, Angew. Chem. 2017, 2018, 2019, Langmuir 2014, 2020, 2021, 2021). All the above papers to understanding of biophysical property of structure of the biomolecules on the membrane are very considerable work for the reviewer and many readers. In this study, Dr. Bankala Krishnarjuna and Prof. Ramamoorthy and co-workers performed the measurement of residual dipolar couplings (RDCs) using negatively charged SMA-EA polymer nanodisc from nucleic acids such as 13C/15N-stable isotope-labeled fluoride riboswitch aptamer. As a consequence, the DMPC- SMA-EA polymer nanodisc exhibited very stable magnetic alignment as shown in 2H spectra. Using this system, they have demonstrated that many obtained RDCs exhibited good correlations. This is interesting per se and suitable for the Biomolecules audience. The reviewer strongly supports the publication of this manuscript without any revision.

Author Response

(The authors gave the same response as above.)
